# Microstructure and Magnetocaloric Effect by Doping C in La-Fe-Si Ribbons

**DOI:** 10.3390/ma15010343

**Published:** 2022-01-04

**Authors:** Huihui Song, Yuhu Hu, Jiale Zhang, Jinyu Fang, Xueling Hou

**Affiliations:** School of Materials Science and Engineering, Shanghai University, Shanghai 200072, China; SONGHUIHUI0731@163.com (H.S.); huyuhu123@shu.edu.cn (Y.H.); zhangjiale907@163.com (J.Z.); fangjinyu0405@163.com (J.F.)

**Keywords:** microstructure, magnetocaloric effect, rapid solidification, annealing, maximum isothermal magnetic entropy

## Abstract

The melt-spun ribbons of LaFe_11.5_Si_1.5_C_x_ (x = 0, 0.1, 0.2, 0.3) compounds are prepared by the melt fast-quenching method. The doping of C is beneficial to the nucleation and precipitation of the La (Fe, Si)_13_ phase, which is indicated by the microstructure observation and the elemental analysis. Subsequently, the ribbons of LaFe_11.5_Si_1.5_C_0.2_ are annealed at different times, and the phase composition, the microstructures, and the magnetic properties are investigated. The LaFe_11.5_Si_1.5_C_0.2_ ribbons annealed at 1273 K for 2 h achieved the best magnetic properties, and the maximum isothermal magnetic entropy change with a value of 9.45 J/(kg·K) upon an applied field of 1.5 T at an increased Curie temperature 255 K.

## 1. Introduction

Magnetic refrigeration, as a new pollution-free and efficient refrigeration technology, has attracted widespread attention and systematic research [1]. Among the magnetic refrigeration materials currently developed, the LaFe_13-x_Si_x_ (1.2 ≤ x ≤ 1.6) alloy is a promising candidate because of its large magnetocaloric effect, low cost, and environmentally friendly properties [1,2,3,4,5,6]. However, there are still some issues, such as the low Curie/working temperature and the long annealing time to generate the La (Fe, Si)_13_ phase producing the large magnetocaloric effect, that hinder this kind of material from practical applications. At present, transition elements such as Co [7,8] and Ni are widely used to replace Fe, or elements such as B [9], H [10], and C [11,12,13] with a small atomic radius can be doped as interstitial atoms to improve the Curie temperature of the La-Fe-Si alloy. The addition of a few rare earth elements, such as Ce instead of La, can greatly improve the magneto-thermal performance of the LaFe_13-x_Si_x_ alloy, but there is the problem of the Curie temperature reduction. Although Co replacing Fe can improve the Curie temperature of the La-Fe-Si alloy, the maximum isothermal magnetic entropy change of the alloy isreduced significantly [14]. Furthermore, the H element is doped as gap atoms in the La-Fe-Si alloy, while the Curie temperature increases, but the hydride is chemically unstable above 330 K, which is an unavoidable problem in practical applications [15].

In 2016, the structural and magnetothermal properties of the LaFe_13-x_Si_x_C_y_ carbide were investigated by V. Paul-Boncour et al. [16], who found that C atom doping leads to an increase in the Curie temperature and a drastic decrease of the magnetic entropy change. An almost single 1:13 phase was obtained after only a 30 min of heat treatment at 1393 K for the ball-milled samples. Even though doping C in the La-Fe-Si alloy ingot can increase the Curie temperature and obtain the optimal magnetic properties with a maximum isothermal magnetic entropy 12.7 J/(kg·K) (ΔH = 5 T) [17,18], the alloy ingot needs a long-time heat treatment for around 1 week for the formation of the La (Fe, Si)_13_ phase. Therefore, in order to increase the Curie temperature and reduce the heat-treatment time simultaneously, we investigated melt-spun ribbons of LaFe_11.5_Si_1.5_C_x_ (x = 0, 0.2) compounds prepared using the melt fast-quenching method. The formation of La (Fe, Si)_13_ phase in solidification and subsequent heat treatments by doping C was studied using an X-ray diffraction analyzer. The magneto-thermal properties were systematically studied using a vibration sample magnetometer.

## 2. Experimental Details

The raw materials used in this experiment were Fe (purity not less than 99.55%), La (purity not less than 99.9%), Si (purity not less than 99.999%), and graphite (purity not less than 99.9%). Considering the volatile rare earth elements in the melting process, the burn loss of the rare earth element La was measured by 10%. To make the sample composition uniform, electromagnetic stirring was initiated during the melting process and each sample was flipped and melted four times. The ingots were melt and spun into ribbons using a melt-spinner with a copper wheel at a surface speed of 35 m/s. For the subsequent heat treatment in a muffle furnace, the melt-spun ribbons were sealed in glass tubes filled with inert gas. The heat-treatments at a temperature of 1273 K to the LaFe_11.5_Si_1.5_C_0.2_ ribbons were 3 min and 2 h, respectively.

The phase structure analysis to the melt-spun ribbons was conducted by an X-ray diffraction instrument, D/MAX-2200-type (Cu target, K_α_-ray). The magnetic properties were determined by a vibration sample magnetometer, namely the Lakeshore7470. The thermal magnetic curve was tested under the 0.1 T magnetic field. The isothermal magnetization curve was tested under the 0–1.5 T magnetic field. The magnetic entropy variation was calculated using the Maxwell Equation (1).
(1)$$\Delta S_{M}\left(T,H\right)=S_{M}\left(T,H\right)-S_{M}\left(T,H=0\right)=\int_{0}^{H}{\left(\frac{\partial M}{\partial T}\right)}_{H}dH$$

## 3. Result and Discussions

### 3.1. Nucleation Rate and Phase Structure

Figure 1a shows the XRD pattern of unannealed LaFe_11.5_Si_1.5_C_x_ (x = 0, 0.1, 0.2, 0.3) ribbons. It is not hard to see from the XRD pattern that in the fast-spun ribbons of LaFe_11.5_Si_1.5_C_x_ (x = 0.2) compounds without heat treatment, the main phases are all α-(Fe, Si) phases, and only a small amount of La (Fe, Si)_13_ phases are contained. With the increase in C content, the relative content of the NaZn_13_ type phase with a magnetocaloric effect increases first and then decreases, and the relative content of the La (Fe, Si)_13_ phase reaches a maximum in the sample of x = 0.2. With the continued increase in C content, the relative content of the La (Fe, Si)_13_ phases tended to decrease. The doping of C favors the formation of the La (Fe, Si)_13_ phases in the LaFe_11.5_Si_1.5_C_x_(x = 0, 0.1, 0.2, and 0.3) alloy. This is because during the rapid solidification process, the La (Fe, Si)_13_ phase competed with the α-(Fe, Si) phase, while the doping of C favored the shaped nucleus and the dissolution of the La (Fe, Si)_13_ phase.

According to the analysis of the jade software, the 2θ of the main peak of the La (Fe, Si)_13_ phase in the unannealed LaFe_11.5_Si_1.5_C_X_ (x = 0, 0.1, 0.2, 0.3) alloy was 46.762°, 46.677°, 46.642°, and 46.512°, respectively, as well as with the doping of the C element. According to the Bragg formula 2dsinθ = λ (d is the interplanar spacing, θ is the diffraction half angle, λ is the wavelength), it can be seen that the interplanar spacing of the La (Fe, Si)_13_ phase in the alloy rapid quenching band was increasing, which shows that C atoms as interstitial atoms entered the lattice of the La (Fe, Si)_13_ phase of the NaZn_13_ cubic structure, which caused the expansion of the crystal structure and the increase of the lattice constant. The results are shown in Table 1.

With the increase in C content, the relative content of the La (Fe, Si)_13_ phase decreased. The doping C was beneficial to the nucleation and precipitation of the La (Fe, Si)_13_ phase in the LaFe_11.5_Si_1.5_C_0.2_ ribbons, because there was a competitive nucleation relationship between the La (Fe, Si)_13_ phase and the α-(Fe, Si) phase during rapid solidification.

The heterogeneous nucleation rate [19,20] can be calculated by the following
(2)$$I=\frac{k_{B}{\mathrm{TN}}_{n}}{3\pi \eta \left(T\right)a_{0}^{3}}\cdot \exp\left[-\frac{\Delta G^{*}}{k_{B}T}\right]$$

Figure 1b,c shows the nucleation rates of the α-(Fe, Si) and La (Fe, Si)_13_ phases versus the under-cooling degree at different C contents. During the solidification process of the ribbons, the degree of under-cooling affects the phase formation mechanism of the La-Fe-Si alloy. Figure 1b shows that in the solidification process of the La-Fe-Si alloy, the nucleation rate of the α-(Fe, Si) phase is higher than that of the La (Fe, Si)_13_ phase when the under-cooling degree is small. Thus, the lower under-cooling degree is not conducive to the formation of the La (Fe, Si)_13_ phase, and the main phase of the alloy is the α-(Fe, Si) phase. When the over cooling degree is large, the shaped nucleus rate of the La (Fe, Si)_13_ phase is higher than the α-(Fe, Si) phase, facilitating the formation of more La (Fe, Si)_13_ phases. The results show that the undercooling degree affects the competitive precipitation of the La (Fe, Si)_13_ phase and α-(Fe, Si) phase.

Under certain chamber pressure, the faster quenching speed, that is, the larger under-cooling degree, creates conditions for the nucleation and precipitation of the La (Fe, Si)_13_ phase, which is beneficial to the effective formation of the La (Fe, Si)_13_ single phase. Secondly, the large undercooling degree during rapid solidification is conducive to the formation of a small La-Fe-Si alloy microstructure [21].

The non-equilibrium rapid solidification process in the La-Fe-Si ribbons provides a high degree of undercooling for the nucleation and precipitation of the La (Fe, Si)_13_ phase, which induces the primary precipitation of the competitive La (Fe, Si)_13_ phase. Meanwhile, the crystal structure of the α-(Fe, Si) phase and La-Fe-Si phase grows slowly, and the nanoscale α-(Fe, Si) phase is distributed periodically and uniformly, which is beneficial to the diffusion of La, Fe, and Si atoms during heat treatment and promotes the inclusion reaction of the La (Fe, Si)_13_ phase. Therefore, the single-phase La (Fe, Si)_13_ phase can be obtained only in a short time by using a fast quenching method to prepare La-Fe-Si alloy rapid quenching strips.

In Figure 2a, region Ⅰ is small, corresponding to the La (Fe, Si)_13_ phase when x = 0, and region Ⅱ is the α-(Fe, Si) phase. The regions between region Ⅰ and region Ⅱ are the transition regions. Figure 2b shows that region Ⅰ (La (Fe, Si)_13_ phase) is significantly increased when x = 0.2. Figure 2c is an enlarged diagram of region Ⅱ, and Figure 2d is an enlarged diagram of the transition region. Table 2 is the EDS analysis of the micro-structure of the La-Fe-Si alloys. With the increasing C content, the content change of each element is not obvious.

### 3.2. LaFe_11.5_Si_1.5_C_0.2_ Heat Treatment

Figure 3 shows the XRD pattern of the LaFe_11.5_Si_1.5_C_0.2_ ribbons annealed at a temperature of 1273 K with different times. As shown in the diagram, the main phase is the α-(Fe, Si) phase and the secondary phase is the La (Fe, Si)_13_ phase to the unannealed LaFe_11.5_Si_1.5_C_0.2_ ribbons. After heat treatment, the main phase changes from the α-(Fe, Si) to the La (Fe, Si)_13_ phase, and the secondary phase changes from the La (Fe, Si)_13_ phase to the α-(Fe, Si) phase. When the heat-treatment time increases from 3 min to 2 h, the relative content of the α-phase decreases. This is due to the inclusion reaction between the α-(Fe, Si) phase and the La (Fe, Si)_13_ phase when the heat treatment of the ribbons is carried out at a temperature of 1273 K for 2 h.

Through the analysis of jade software, it is found that with the extension of heat treatment time, the 2θ of the main peak of the 13 phases of La (Fe, Si) in the fast quenched strip are 46.800°, 46.730°, and 46.698°, respectively. The main peak of La (Fe, Si)_13_ phase shifts to a small angle, because with the increase in heat treatment time, C atoms are fully spaced from the lattice of the La (Fe, Si)_13_ phase with a NaZn_13_ cubic structure, which makes its lattice expand and causes the lattice constant of the La (Fe, Si)_13_ phase to increase [22]. In addition, it can be clearly seen from the figure that the La (Fe, Si)_13_ phase in the alloy has become the main phase when the LaFe_11.5_Si_1.5_C_0.2_ strip is heat treated for 3 min. Compared with the alloy samples prepared by the traditional melting ingot method, the melt quenching process with a certain rapid quenching speed provides deep undercooling conditions for the formation of the La (Fe, Si)_13_ phase in the peritectic reaction process. It promotes the competitive nucleation and precipitation of La (Fe, Si)_13_ phase in the rapid solidification process, and the La (Fe, Si)_13_ phase formed in the early stage and the refined La (Fe, Si)_13_ phase grains in the melt rapid quenching shorten the time required for inclusion reaction in the heat treatment process. The results in Table 3 show that the lattice parameters are 11.4985 Å, 11.5036 Å, and 11.5107 Å at a wheel speed of 35 m/s annealed at 1273 K for as spun, 3 min and 2 h, respectively. In other words, the longer annealing time, the bigger the expansion of the alloy lattice.

Figure 4 is a free surface SEM appearance of the LaFe_11.5_Si_1.5_C_0.2_ fast quenching strip at 35 m/s at different times at a temperature of 1273 K. As is seen in Figure 4a, the free surface of the rapid quenching ribbon without heat treatment has a flat surface, no obvious branch crystal tissue, with cluster boundaries similar to the crystal boundary, probably due to the fast cooling speed and small grain size. After 3 min of heat treatment, some white particles of the rapid quenching ribbon began to precipitate through the EDS analysis (see Table 4). After the preliminary analysis of the research group, it can be inferred that the white particles are La_2_O_3_ [23]. Through the energy spectrum analysis of the free surface grain after heat treatment (see Table 4), in the atomic percentage of each element at different times, the internal phase composition of the grain is close to the La (Fe, Si)_13_ phase, and the analysis results are consistent with the results of the XRD in Figure 3. As can be seen from Figure 4c, in the free surface of the ribbon after heat treatment for 2 h, the triangle appearance has grown almost completely into a quadrilateral appearance, has spread over the whole surface, the crystal boundary is relatively flat, and the white particles of the La-rich phase are mostly distributed at the grain boundary of the circle particles, rarely at the quadrilateral crystal boundary.

Figure 5 shows the microstructure appearance image and high resolution image of LaFe_11.5_Si_1.5_C_0.2_ with a rapid quenching speed of 35 m/s and 3 min of heat treatment. Figure 5b is the Fourier transform of the lattice stripe of the circle region of Figure 5a, calibrated as the uniform La (Fe, Si)_13_ phase. Figure 5c is the Fourier transform of the lattice stripes of the Figure 5d white strips, labeled as a uniform La-Fe-Si phase. Figure 5e is the Fourier transform of the lattice stripe of the Figure 5a dark area, labeled as a uniform α-(Fe, Si) phase.

Figure 6 shows the tissue appearance image and high resolution image of LaFe_11.5_Si_1.5_C_0.2_ with a fast quenching speed of 35 m/s and 2 h heat treatment. As can be seen from Figure 6a, the area in the fast quenching strip consists of two different shapes. Figure 6b is an enlarged picture of region 1. It can be found that the quadular bulge in the SEM diagram of the free surface is composed of small and uniform particles, the matrix consists of gray and white particles, with a particle size within 200–500nm, and the two shapes are distinguished by a straight boundary. The formation of a quadratic crystal boundary in Figure 4c is also confirmed.

From the LaFe_11.5_Si_1.5_C_0.__2_ fast quenched strip powder XRD pattern with a fast quenching speed of 35 m/s in Figure 3, the main phase is the La (Fe, Si)_13_ phase, containing only a small number of α-(Fe, Si) phases, so a large number of quadrilateral bumps in the free surface should be a relatively uniform La (Fe, Si)_13_ phase and a small number of α-(Fe, Si) phases in the white particles. A high-resolution morphology is taken at the junction of the base and the quadrilateral projection, as shown in Figure 6c. The Fourier transform of the lattice stripes of the high-resolution matrix A region is normalized to the uniform La (Fe, Si)_13_ phase, the lattice stripe of the gray grain in the B, C region of a high resolution, and the raised gray particles to the La-Fe-Si phase.

### 3.3. Effects on the Magnetic Properties

As can be seen from Figure 7, with the extension of the heat treatment time, the Curie temperature of the LaFe_11.5_Si_1.5_C_0.2_ quenching strip increases at 224 K (0 min), 231 K (3 min), and 255 K (2 h). This is because, with the increase of heat treatment time, the lattice expansion of the NaZn_13_ structure is caused by the effective entry of atomic energy into the gap position of La (Fe, Si)_13_ phase C. The three strong peaks of the La (Fe, Si)_13_ phase in Figure 3 can effectively prove this. With the C atoms entering the gap position in the La (Fe, Si)_13_ lattice, the 3D band of the Fe becomes narrower, the ferromagnetic interaction is enhanced, and the curie temperature tends to increase obviously.

Figure 8 shows the LaFe_11.5_Si_1.5_C_0.2_ fast quenching strip with a fast quenching speed of 35 m/s at a temperature of 1273 K, the maximum isothermal temperature after different times of heat treatment. We can see from Figure 8 that when the heat treatment time is 0 min, 3 min, and 2 h, the maximum isothermal magnetic entropy change of the LaFe_11.5_Si_1.5_C_0.2_ fast quenched strip is 2.32 J/(kg·K), 6.8 J/(kg·K), and 9.45 J/(kg·K), respectively. The maximum isothermal magnetic entropy change mutated after 20 min of heat treatment, and then showed an obvious trend of first increasing and then decreasing, and reached the maximum value after 2 h of heat treatment. This change in the magneto-thermal effect as the heat treatment time extends comes from the following reason. It is difficult to complete the crystallization reaction of the La (Fe, Si)_13_ phase during solidification, and La-Fe-Si as the primary α-(Fe, Si) phase is the main phase in the fast-quenched strip of the alloy, and the relative content of the La (Fe, Si)_13_ phase with a giant magnetothermic effect is relatively small, so it has a small maximum isothermal magnetic entropy change. After thermal treatment, during the wafer coating reaction process, the not fully reactive α-(Fe, Si) phase and the La-Fe-Si phase generates the La (Fe, Si)_13_ phase, causing the La (Fe, Si)_13_ phase in the alloy, thus having a large maximum isothermal magnetic entropy change, and mutations for the slightly longer thermal treatment (20 min). This agrees with the XRD result in Figure 3.

Figure 9 is the 3D curve of the temperature, magnetic field, and maximum isothermal magnetic entropy change of (x = 0.2) after thermal treatment for 2 h at 1273 K. As the magnetic fields increase, the ΔS−T curve changes from the symmetrical herringbone to the asymmetric curve, indicating that the alloy phase transition type from the secondary phase transition to the primary phase transition and ΔS shows an increasing trend (because the primary phase transition is the change of material magnetic ordered state caused by lattice distortion, the resulting magnetic entropy change is much greater than the secondary phase transition and reaches values of 9.45 J/(kg·K) upon an applied field of 1.5 T).

## 4. Conclusions

Considering the disadvantages of the low magneto-thermal effect and the long heat treatment time of room temperature magnetic refrigeration materials using the La-Fe-Si alloy, the magneto-thermal effect is improved, and the heat treatment time in the preparation process is greatly shortened by the melt fast quenching process. At the same time, the effects of different heat treatment times on the phase composition, magnetic properties, and micro-tissue of LaFe_11.5_Si_1.5_C_0.2_ are also studied. We present the following conclusions:The doping of C promotes the formation of La (Fe, Si)_13_ phases in the La-Fe-Si series alloy. Compared with La-Fe-Si alloy without C doping, the LaFe_11.5_Si_1.5_C_x_ (x = 0.1, 0.2, 0.3) alloy obtained more of the La (Fe, Si)_13_ phase without heat treatment.The process of heat treatment for 2 h at 1273 K facilitates a large isothermal variation of LaFe_11.5_Si_1.5_C_0.2_ entropy of alloy. With the extended thermal treatment time, the maximum isothermal magnetic entropy change of the LaFe_11.5_Si_1.5_C_0.2_ alloy fast strip tends to increase first before decreasing, reaching a maximum at 2 h of thermal treatment of 9.45 J/(kg·K).The characteristic quadrangle morphology in the LaFe_11.5_Si_1.5_C_0.2_ alloy fast quenching strip with 2 h pf heat treatment is benefitted by obtaining a higher magneto-thermal effect. Through the transmission analysis, the quadrilateral convex appearance in the 2 h heat treatment is the uniformly distributed La (Fe, Si)_13_ phase, and also the uniformly staggered distributed α-(Fe, Si) phase in the fast quenching band and the La-Fe-Si phase, which facilitates the contact between the α-(Fe, Si) phase and the La-Fe-Si phase, and promotes the packet analysis reaction. The uneven α-(Fe, Si) phase white large particles distributed in the alloy strip during 3 min heat treatment are difficult to contact using La-Fe-Si during heat treatment, which is not conducive to the packet analysis reaction, so the magneto-thermal effect is poor.

## Figures and Tables

**Figure 1 materials-15-00343-f001:**
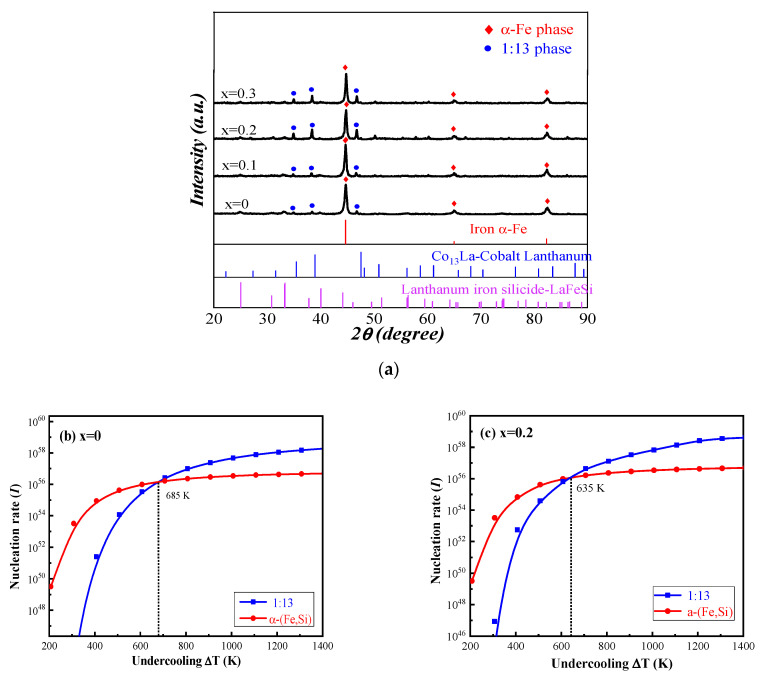
X-ray diffraction patterns of unannealed LaFe_11.5_Si_1.5_C_x_ (x = 0, 0.1, 0.2, 0.3) ribbons at a surface speed of 35 m/s (**a**); calculated nucleation rates of the α-(Fe, Si) and La (Fe, Si)_13_ phases versus under-cooling degrees at different C contents; (**b**) x = 0; (**c**) x = 0.2.

**Figure 2 materials-15-00343-f002:**
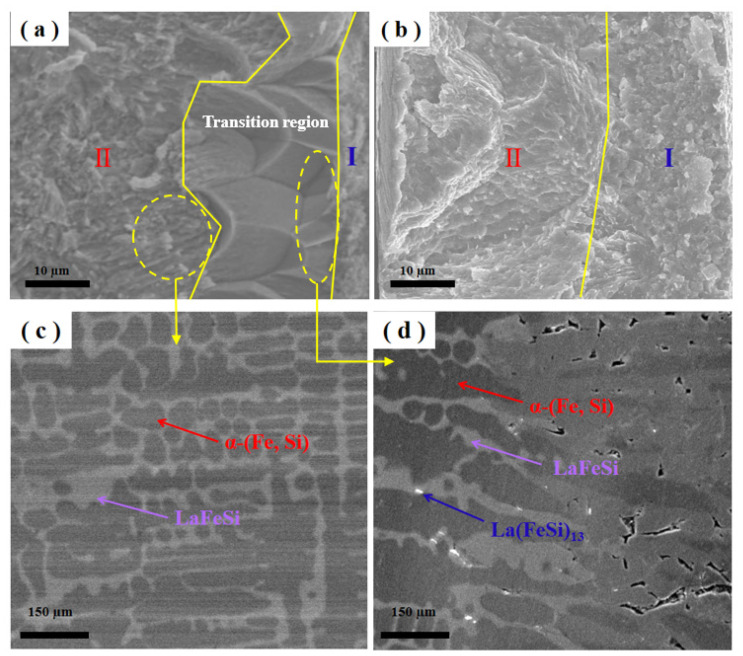
SEM images of a melt-spun La-Fe-Si ribbon (**a**,**b**) cross-sectional images of a melt-spun LaFe_11.5_Si_1.5_C_x_ (x = 0, 0.2) ribbons; (**c**) Magnification of region Ⅱ; (**d**) Magnification of the transition region.

**Figure 3 materials-15-00343-f003:**
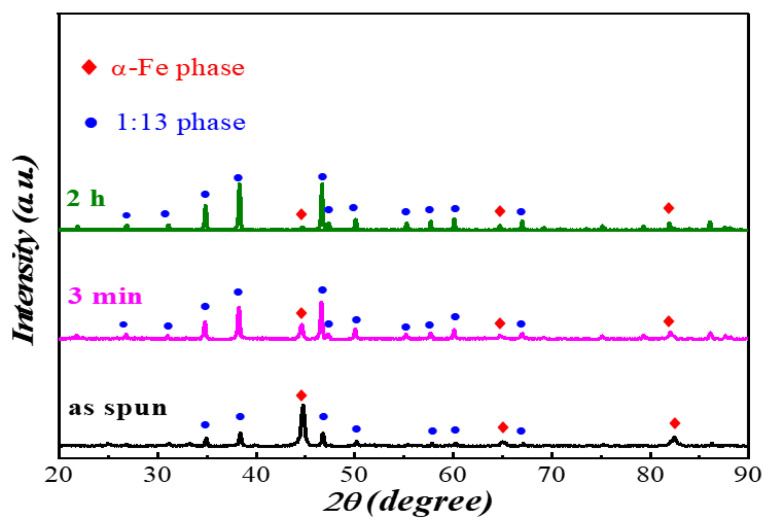
X-ray diffraction patterns of LaFe_11.5_Si_1.5_C_0.2_ ribbons at a wheel speed of 35 m/s annealed at 1273 K for different times.

**Figure 4 materials-15-00343-f004:**
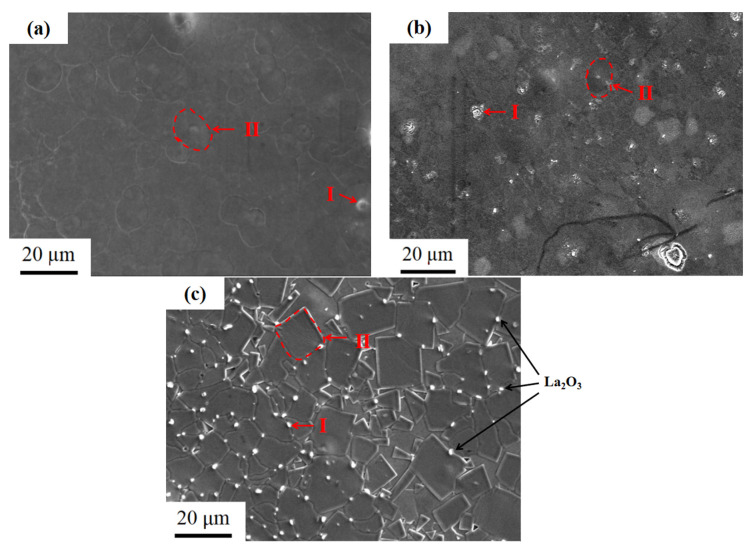
SEM images of LaFe_11.5_Si_1.5_C_0.2_ ribbons annealed at 1273 K for different times: (**a**) 0 min; (**b**) 3 min; (**c**) 2 h.

**Figure 5 materials-15-00343-f005:**
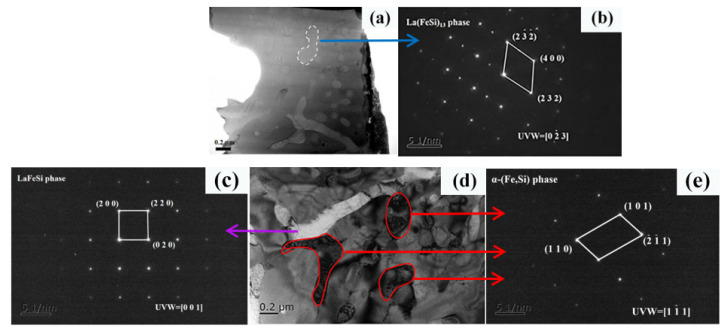
Selected area electron diffraction of LaFe_11.5_Si_1.5_C_0.2_ ribbons after heat treatment for 3 min. (**a**) Display of selected area in TEM; (**b**) Fourier transform at the SAED of (**a**); (**c**) Fourier transform at the SAED of (**d**) white strips; (**d**) display of selected area in TEM; (**e**) Fourier transform at the SAED of (**a**–**e**).

**Figure 6 materials-15-00343-f006:**
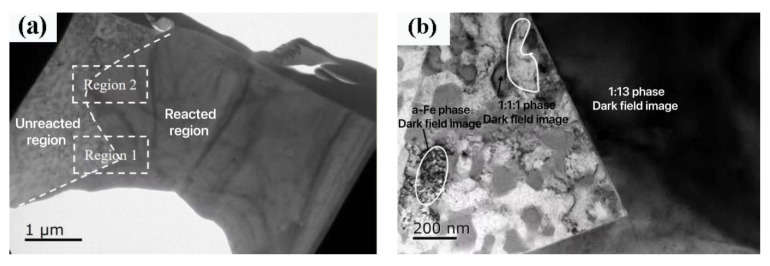
(**a**,**b**) Microstructure morphology (**c**,**f**) HRTEM micro-graph; (**d**) Fourier transform; (**e**) display of selected area in TEM of 35 m/s LaFe_11.5_Si_1.5_C_0.2_ ribbons near the free surface annealed at 1273 K for 2 h.

**Figure 7 materials-15-00343-f007:**
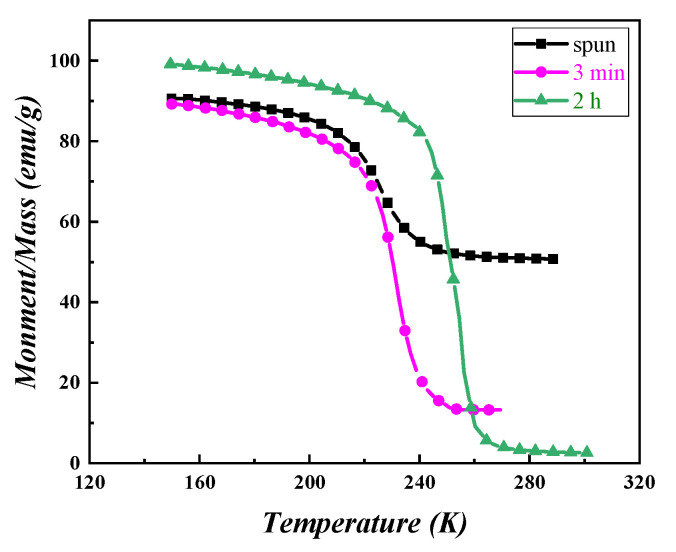
Thermomagnetic curves of 35 m/s LaFe_11.5_Si_1.5_C_0.2_ ribbons annealed at 1273 K for different times.

**Figure 8 materials-15-00343-f008:**
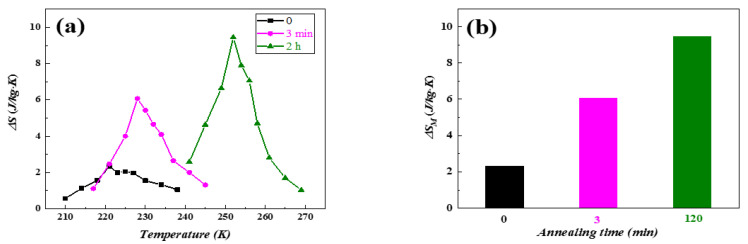
(**a**) ΔS−T curves and (**b**) histogram of the maximum isothermal magnetic entropy of LaFe_11.5_Si_1.5_C_0.2_ ribbons at a wheel speed of 35 m/s annealed at 1273 K for different times.

**Figure 9 materials-15-00343-f009:**
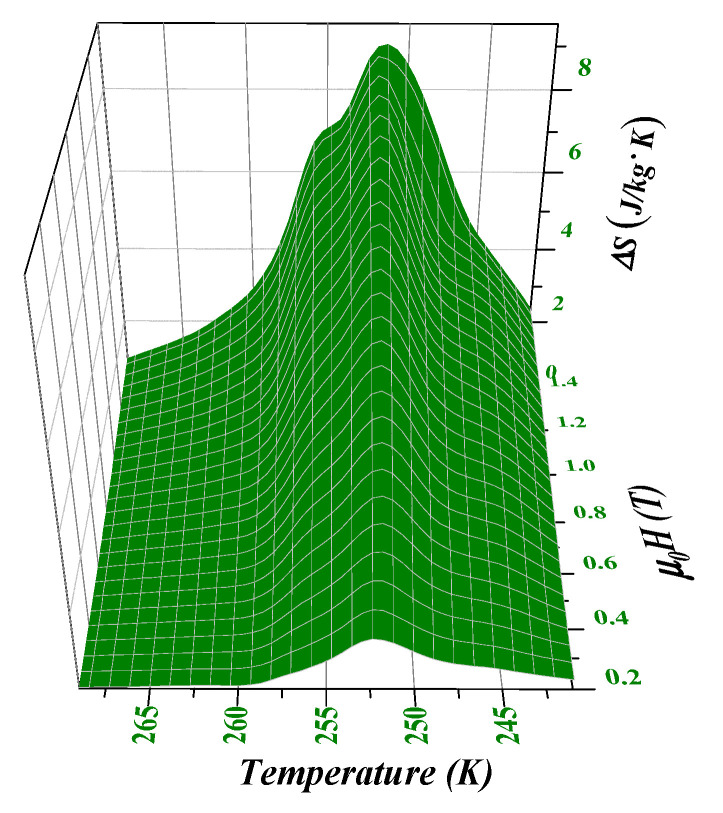
3D curve of the temperature, magnetic field, and maximum isothermal magnetic entropy variation of the LaFe_11.5_Si_1.5_C_0.2_ fast quenched band at 35 m/s (after 1273 K × 2 h heat treatment).

**Table 1 materials-15-00343-t001:** Lattice parameters of unannealed LaFe_11.5_Si_1.5_C_x_ (x = 0, 0.1, 0.2, 0.3) ribbons.

C Content	Lattice Parameters (Å)
x = 0	11.4883
x = 0.1	11.4931
x = 0.2	11.4985
x = 0.3	11.5024

**Table 2 materials-15-00343-t002:** The EDS analysis of the micro-structure of LaFe_11.5_Si_1.5_C_x_ (x = 0, 0.2) alloys.

Chemical Composition		La (at %)	Fe (at %)	Si (at %)	Phase
x = 0	white point	14.28	70.58	15.14	La (Fe, Si)_13_
	dark gray	0	94.48	5.52	α-(Fe, Si)
	gray white	34.26	32.65	33.09	LaFeSi
x = 0.2	white point	13.87	71.14	14.99	La (Fe, Si)_13_
	dark gray	0	94.45	5.55	α-(Fe, Si)
	gray white	33.64	32.97	33.39	LaFeSi

**Table 3 materials-15-00343-t003:** Lattice parameters of annealed LaFe_11.5_Si_1.5_C_0.2_ ribbons at 1273 K for different times.

Annealing Time	Lattice Parameters (Å)
as spun	11.4985
3 min	11.5036
2 h	11.5107

**Table 4 materials-15-00343-t004:** Inside grains and white grains the EDS analysis of LaFe_11.5_Si_1.5_C_0.2_ ribbons at a wheel speed of 35 m/s annealed at 1273 K for different times.

Heat Treatment Time	Area	La (at%)	Fe (at%)	Si (at%)	O (at%)	C (at%)	Phase
0 min	I (white particles)	9.11	74.82	7.43	4.97	3.66	La_2_O_3_
	II (intracrystalline)	11.84	72.56	10.94	0.08	4.58	La (Fe, Si)_13_
3 min	I (white particles)	15.28	49.74	6.36	21.63	6.99	La_2_O_3_
	II (intracrystalline)	11.73	71.62	11.01	1.33	4.31	La (Fe, Si)_13_
2 h	I (white particles)	18.97	25.93	4.96	36.88	13.26	La_2_O_3_
	II (intracrystalline)	11.33	69.10	11.14	4.37	4.06	La (Fe, Si)_13_

## Data Availability

The study does not include publicly archived datasets.

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
