# Peer review of "Microstructure and Magnetocaloric Effect by Doping C in La-Fe-Si Ribbons"

_materials, 2022, doi:10.3390/ma15010343_

Round 1

Reviewer 1 Report

In this work, the authors have prepared ribbons of LaFe11.5Si1.5Cx (x=0, 0.2) compounds by the melt fast-quenching method. The formation of the La(Fe, Si)13 phase have achieved within a short time. Further The microstructure and magnetocaloric effect were studied with doping of C. The manuscript is well written and can be accepted for possible publication in this journal after the following corrections.

  1. Abstract needs to be modify properly. “The surface speed of the spinning wheel is 35 m/s” this sentence should be a part of experimental section.
  2. Proper indexing of each plane is required in XRD with an explanation of lattice parameters and crystal symmetry.
  3. Authors studied the magnetothermal properties of LaFe11.5Si1.5Cx (x=0, 0.2) the highest properties are achieved when the content of C is 0.2. what about further increase in the C beyond 0.2.

Reviewer 2 Report

Authors showed how Miscrostructure and magnetocaloric effect changes by doping C in La-Fe-Si ribbons.

  1. Interestingly that such effect of C on La-Fe-Si already was established for the bulk materials and authors should refer to the research from V. Paul-Boncour 's group. one as an example https://doi.org/10.1016/j.jssc.2015.10.016
  2. some features in the introduction should be properly cited even it is well known.
  3. it is already known that with ball milling , spark plasma sintering etc you can reduce annealing time of LaFeSi (1-13) alloys to a couple of hours . Please revisit literature.
  4. in the text and even in the table 1 presented information only about LaFeSi (1-13) and a-FeSi but nothing LaFeSi phase which obviously seen in the fig 2.
  5. Why La2O3 is not visible on the XRD?
  6. Fig6 please use english notes.
  7. 'the lattice expansion of NaZn13 structure' would be nice to see the numbers from the XRD rietveld refinement. also would be good to present amount of phases during each time of annaeling and at the beginning.
  8. conclusion about C effect should be rewritten since it is already known effect.

Round 2

Reviewer 2 Report

Jade software or jade 6.0- Please, stick to one possible abbreviation
